# Serological Screening for Antibodies against SARS-CoV-2 in Dutch Shelter Cats

**DOI:** 10.3390/v13081634

**Published:** 2021-08-18

**Authors:** W. J. R. van der Leij, Els M. Broens, Jan Willem Hesselink, Nancy Schuurman, Johannes C. M. Vernooij, Herman F. Egberink

**Affiliations:** 1Department of Clinical Sciences, Shelter Medicine Program, Faculty of Veterinary Medicine, Utrecht University, Yalelaan 108, 3584 CM Utrecht, The Netherlands; 2Veterinary Microbiological Diagnostic Centre (VMDC), Department of Biomolecular Health Sciences, Faculty of Veterinary Medicine, Utrecht University, Yalelaan 1, 3584 CL Utrecht, The Netherlands; E.M.Broens@uu.nl; 3Department of Clinical Sciences, Faculty of Veterinary Medicine, Utrecht University, Yalelaan 108, 3584 CM Utrecht, The Netherlands; J.W.Hesselink@uu.nl; 4Virology Division, Department of Biomolecular Health Sciences, Faculty of Veterinary Medicine, Utrecht University, Yalelaan 1, 3584 CL Utrecht, The Netherlands; N.M.P.Schuurman@uu.nl (N.S.); H.F.Egberink@uu.nl (H.F.E.); 5Department of Population Health Sciences, Faculty of Veterinary Medicine, Utrecht University, Yalelaan 7, 3584 CL Utrecht, The Netherlands; J.C.M.Vernooij@uu.nl

**Keywords:** Coronavirus disease 2019 (COVID-19), neutralizing SARS-CoV-2 antibody, seroprevalence, serology, titer, animal shelter, one health, shelter medicine

## Abstract

The COVID-19 pandemic raised concerns that companion animals might be infected with, and could become a reservoir of, SARS-CoV-2. As cats are popular pets and susceptible to Coronavirus, we investigated the seroprevalence of SARS-CoV-2 antibodies in shelter cats housed in Dutch animal shelters during the COVID-19 pandemic. In this large-scale cross-sectional study, serum samples of shelter cats were collected during the second wave of human COVID-19 infections in The Netherlands. Seroprevalence was determined by using an indirect protein-based ELISA validated for cats, and a Virus Neutralization Test (VNT) as confirmation. To screen for feline SARS-CoV-2 shedding, oropharyngeal and rectal swabs of cats positive for ELISA and/or VNT were analyzed using PCR tests. In 28 Dutch animal shelters, 240 shelter cats were convenience sampled. Two of these cats (0.8%; CI 95%: 0.1–3.0%) were seropositive, as evidenced by the presence of SARS-CoV-2 neutralizing antibodies. The seropositive animals tested PCR negative for SARS-CoV-2. Based on the results of this study, it is unlikely that shelter cats could be a reservoir of SARS-CoV-2 or pose a (significant) risk to public health.

## 1. Introduction

The outbreak of SARS-CoV-2 infections among the human population in the Chinese city of Wuhan in December 2019 was the start of a pandemic, which spread quickly around the world by transmission via close contact between people. Several non-human animal species are also susceptible to infection with SARS-CoV-2, including cats, dogs, and mink. Infections in cats were shown under experimental [1,2] and natural living conditions [3,4]. From the start of the pandemic, feline cases of naturally SARS-CoV-2-infected individual domestic cats were reported in several countries, such as Hong Kong, Japan, and Russia; in Europe: Belgium, France, Spain, and Germany; the USA, Canada; in South America: Chile and Brazil [5]. The ability of the virus to transmit between cats [6,7] was demonstrated after experimental infection found feline susceptibility to be comparable with SARS-CoV infection in 2003 [8].

The risk of mammalian SARS-CoV-2 reservoirs became clear when Dutch mink farm workers contracted COVID-19, most likely by inhaling contaminated dust from SARS-CoV-2-infected minks [9,10].

Early in the outbreak, concerns were raised about the possible role of cats in the dynamics of this infectious disease. Cats from owners diagnosed with COVID-19 tested positive for SARS-CoV-2 [11]; these cats were most likely infected by their owners. In addition, there was also the fear of transmission from cats to humans (as feline cells share the same SARS-CoV-2 receptor with humans—the angiotensin-converting enzyme II (ACE2) [12]). The possible transmission of SARS-CoV-2 from cats to humans caused concern among cat owners and animal shelter staff [13], not only because of the potential risk for zoonotic transmission, but also because of the anticipated public aversion to (stray) cats [14,15,16].

Billions of cats worldwide live in close contact with their owners and millions of cats pass through temporal stays in animal shelters. Zoonotic pathogens do emerge in shelters [17,18]. Within a One Health approach, shelters can be sentinels for these emerging diseases. Many shelter cats are communally housed, thus facilitating the transmission of pathogens. Although there is no evidence to date that cats play an important role in the epidemiology of SARS-CoV-2 and its possible transmission to humans, it seems important to monitor the population of shelter cats to determine the SARS-CoV-2 antibody prevalence in the shelter environment, and to estimate the possibility of transmission among shelter cats and/or the animal–human spillover of (new mutants of) SARS-CoV-2. The objective of this study was to determine the seroprevalence of SARS-CoV-2 in Dutch shelter cats.

## 2. Materials and Methods

### 2.1. Animals, Data Collection and Sampling

Forty private animal shelters were contacted. The aim was to include three shelters per province (12 provinces in total) in The Netherlands. In the two southern provinces, Noord-Brabant and Limburg, 12 shelters were invited because numerous outbreaks of SARS-CoV-2 in humans and farmed minks [9] occurred during the start of the pandemic in February 2020 in these two regions. Each shelter was asked to send in serum samples from 10 shelter cats meeting the following inclusion criteria:-Age: half a year or older (to avoid possible detection of maternal antibodies against SARS-CoV-2 in young cats);-Living conditions in the shelter: preferably group housed cats or cats soon entering group housing (within two days after sampling);-Chances for rehoming: cats should be sociable with humans and available for adoption (as potential spillovers of SARS-CoV-2 to new cat owners).

Informed consent was requested of every participating shelter. Information about the management of the facility and its housing conditions was collected by means of a questionnaire (Appendix B). Individual information about the gender, origin, age, and health status of every cat included in the study was gathered through a detailed survey.

All shelters were provided with sampling materials to facilitate sampling in a uniform way. Convenience samples of serum were taken from the cats that met the inclusion criteria by the shelter veterinarian.

In most cases, one serum sample per cat was taken for analysis. When more samples per cat were taken and analyzed, only the first samples were used for analysis.

### 2.2. Serology

Neutralizing antibodies against Coronaviruses target the S protein. This protein consists of two subunits, S1 and S2, of which S1 contains the presumed Receptor Binding Domain (RBD).

Antibody detection was performed using assays, as described by Zhao et al. [19]. In short, ELISA plates were coated with SARS-CoV-2 S1 and RBD proteins diluted in phosphate-buffered saline, and blocked with blocking buffer. Sera were screened at a dilution of 1:100 using a goat anti-cat IgG horseradish peroxidase (HRP) as a secondary antibody (Rockland Immunochemicals, Inc., https://rockland-inc.com, accessed on 5 August 2021). The optical densities (ODs) in the ELISAs were measured at 450 nm and cut-off values were determined at six times the Standard Deviations above the mean value of reactivity of seronegative samples from a pre-COVID-19 cohort. Every positive ELISA result for one or both SARS-CoV-2 proteins (S1 and RBD) was confirmed with a VNT. Interpretation of the results was as in Zhao et al. [19]. ‘Seropositive’ samples were defined as any sample that was ELISA-positive for both SARS-CoV-2 proteins (S1 and RBD), combined with a virus neutralization titer (VNT) ≥ 16. Samples that were ELISA positive for both S1 and RBD, but with a negative VNT, were considered ‘suspected’. Samples that were positive for only one of the viral proteins and a negative VNT were considered negative.

To assess the risk of seropositive cats shedding SARS-CoV-2, oropharyngeal and rectal samples were taken from the cats immediately after they proved to be seropositive. The purpose of taking these samples was to protect the safety of shelter staff by assuring that serological positive cats were not shedding SARS-CoV-2. The swabs were analyzed with a real-time RT-PCR assay, using the same probes and primers as previously described by Corman et al. [20].

### 2.3. Statistics

Results from the serology analyses, registered information per individual cat, and answers to the questionnaires were entered into a Microsoft Excel spreadsheet. The proportion of seropositive samples (seroprevalence) was calculated at a 95% confidence interval, based on a binomial distribution, using the web-based Free Statistics Calculators [21].

## 3. Results

### 3.1. Animal Shelters

Of the 40 invited animal shelters, 28 agreed to participate, including seven shelters in the southern provinces of Noord-Brabant and Limburg. The distribution of the Dutch municipalities served by the participating animal shelters is shown in Figure 1 and in Appendix A.

Only one shelter exclusively facilitated the intake and rehoming of cats; the other shelters accepted both cats and dogs, while 11 shelters accepted other (stray) pet species (e.g., rabbits, rodents, ferrets, domestic birds) as well.

### 3.2. Characteristics of the Study Population

In total, serum samples of 251 cats were retrieved from August 2020 to February 2021. Samples of 11 cats were excluded: nine due to age (too young at time of sampling: <0.5 year), one for lacking information about age and bodyweight, and one due to the quality of the serum sample. For descriptive purposes, four age categories were used, in line with Rix et al. [22]: kittens (0.5–<1 year), young adults (1–<4 years), mature adults (4–<7 years) and seniors (≥7 years) (Table 1).

The median age of 238 cats (as the specific age of two adult cats was lacking) was 3.0 years (range 0.5–16.5 years). Two thirds (66%) of these 240 cats came in as strays, while the rest (31%) were surrendered by their owners or confiscated (3%) due to animal welfare issues. The Length of Stay (LOS) at sampling (mean: 56.6; median = 23.0; range = 0–1833 days) was ≤14 days for 21% of the cats, while no date of entry was known for five cats (2%). Two thirds (63%) of the cats lived in social groups when sampled, while 35% were still housed in solitary cages.

### 3.3. Sampling

Serum samples of 240 cats were incorporated into this study. Of the 28 participating shelters, 2 shelters provided more than 10 samples, 16 shelters submitted 10 samples, and the remaining 10 shelters provided 8 samples or less (range: 2–8 samples). The mean number of samples per shelter was 8.5 samples/shelter. At sampling, the health of the cats was evaluated. Of 237 cats, 49 showed clinical signs, including 17 cats with upper respiratory disease (URD), 8 cats with gastrointestinal problems, and 24 with other medical conditions (Appendix A). No data were available for 3 of the 240 cats.

### 3.4. Intake of Pets from COVID-19 Related Households

Almost one third of the shelters (8/27; one shelter did not answer) reported the intake of pets from COVID-19 infected households, including cats. Two of these cats were included in this seroprevalence study, while the rest were rehomed before this study started. In one case, the owner had recently passed away because of COVID-19. This cat was tested at shelter intake for SARS-CoV-2 antibodies and by PCR testing. Its SARS-CoV-2 antibody titer proved to be negative, as were the oropharyngeal and rectal swabs of this cat. Another owner surrendered a cat in October 2020, having been infected with COVID-19 in March 2020. This cat was also seronegative (no PCR test was carried out) at shelter intake.

### 3.5. Serology

In our study, we used the S1 and RBD ELISA, as published by Zhao et al. [19]. In their study, they showed that there was no cross-reactivity between the S1 and RBD of the FCoV. Additionally, there was no cross-reaction in the VNT, corroborating the specificity of the assays used in this study for SARS-CoV-2. A total of 240 cat samples were tested using the ELISAs with SARS-CoV-2 proteins S1 and RBD. Five cats showed positive ELISA results for both proteins (S1 and RBD), and five cats had positive ELISAs for either S1 or RBD. These 10 feline sera were then screened with the Virus Neutralisation Test (VNT). Two of these ten samples showed a positive VNT and were scored ‘seropositive’ according to the criteria previously defined by Zhao et al. [19] (i.e., ELISA-positive for both S1 and RBD, plus a positive VNT), resulting in a seroprevalence of 0.8% (95% CI: 0.1–3.0%). Three out of these ten cats were ELISA-positive for both S1 and RBD, but negative in the VNT, and were, therefore, scored as ‘suspected’. When including the suspected samples, a seroprevalence of 2.1% (5/240, 95% CI: 0.7–4.8%) was determined. The remaining five samples (ELISA-positive for only one of the two proteins) were VNT-negative and considered ‘seronegative’. The other 230 cat samples were ELISA-negative for both proteins.

The two SARS-CoV-2-antibody-positive cats were housed in two different shelters in different parts of the country. The first seropositive cat (two samples: 1st sample on 18th of September 2020, 2nd sample 33 days later) had VNT-titers of 128 and 512 (sample one and two, respectively). This was a female cat (castrated, BW 2.8 kg, eight years old, no clinical signs) that was surrendered to a shelter in the province of Limburg by its owners as a 100% indoor cat. The owner confirmed to the shelter that they were not a COVID-19-patient. After an initial quarantine period of 14 days, this cat was socially housed with 11 other cats. It stayed in group housing for 80 days before being sampled for this study. During this period, the shelter added five other cats to this group (with a range of co-housing between 35 and 56 days; mean: 42 days) which were included in this study. The shelter provided 10 samples for this study and none of the other cats (including these five group mates) proved to be seropositive.

The second seropositive cat (sampled on 22 December 2020) had a VNT-titer of >512. This was a male cat (castrated, BW 2.9 kg, low body condition score, clinical signs of pancreatitis) that came in as a stray from the province of Zuid Holland, with an estimated age of 15 years. This cat was in quarantine for only three days before being sampled. This shelter provided seven samples for this study. No other cat was found to be seropositive in this shelter.

### 3.6. PCR SARS-CoV-2

Of all 10 cats with ELISA inconclusive/positive results, oropharyngeal and rectal swabs were taken and tested by PCR. None of them tested as PCR positive, indicating that these 10 shelter cats were not shedding the virus at the time of sampling.

## 4. Discussion

In this study, we found that five out of 240 cats (2.1%) were ELISA-positive for both viral proteins, S1 and RBD. Two of the samples were also positive for VNT, giving a seroprevalence of 0.8% against SARS-CoV-2, based on the criteria defined in the study by Zhao et al. Based on the results of this study, it is unlikely that animal shelters create a potential spillover of SARS-CoV-2 to shelter cats and, consequently, to humans.

The reasons for the inconsistency between the ELISA double-positive samples and their negative VNT results, however, are unknown; more studies are needed to elucidate the kinetics of feline SARS-CoV-2 antibodies after infection.

In cats living under non-experimental circumstances such as pets, stray cats, and shelter cats, naturally occurring COVID-19 infections have been reported. Zhang et al. [4] were the first to investigate feline infections by SARS-CoV-2 in the city of Wuhan. The sampled feline population consisted of 102 cats in total, coming from different sources such as pet hospitals, stray/shelter cats, and pets from COVID-19 patients. Fifteen samples proved to be ELISA positive, including six stray cats initially taken in by animal shelters after the outbreak. However, Zhang’s study did not specify the number of cats per source, meaning that the seroprevalence among the Wuhan stray/shelter cats remained unknown.

Some European studies have reported on SARS-CoV-2 seroprevalences in cats. Within a population of 105 Italian cats, including stray and shelter cats, a seroprevalence of 1% was reported [23], while in a population of 114 Spanish stray cats, a 3.5% seroreactivity to SARS-CoV-2 [24] was found. The differences in seroprevalence could be due to multiple factors. In the Spanish study, cats were captured in one location (the city of Zaragoza), whereas cats in the Italian study were living in a high endemic area in Northern Italy, with a higher prevalence of SARS-CoV-2 infections in humans compared to the different areas in The Netherlands [25].

Other factors could also have contributed to the differences in seroprevalence between the studies. The diagnostic assays used in the current study varied from those used in the Italian and Spanish studies. Villaneuva-Saz et al. [24] used the RBD protein in their ELISA without VNT confirmation, while Spada et al. [23] used the viral nucleocapsid N protein. Zhao et al. reported that eight out of nine SARS-CoV-2-free cats infected with feline corona virus (FCoV) showed an antibody reacting with the SARS-CoV-2 N protein. This is likely caused by antigenic cross-reactivity between the N proteins of SARS-CoV-2 and feline corona virus (FCoV), which led to disqualification of the N protein as an appropriate antigen for the serologic screening of samples from the cats [19].

In the current study, the two seropositive cats were older cats (a female of eight years and a male of approximately 15 years of age), of which the female showed no clinical signs, while the male was reported to show clinical signs of pancreatitis and had a low body condition score. In clinical studies [2,6], no pancreatic pathology has been reported in COVID-19 infected cats. In human patients, however, an association was described between acute pancreatitis and COVID-19, though without evidence of causality between SARS-CoV-2 and the infection [26].

During the 7-month period of sampling cats for this study, The Netherlands experienced a second wave of human SARS-CoV-2 infections, with over a tenfold increase in human daily infections [27], and provinces such as Zuid Holland and Limburg were relatively more affected [28].

The seropositive female cat had been surrendered to an animal shelter as an indoor cat, by an owner who was not infected with COVID-19. It is unknown where the cat became infected with SARS-CoV-2. As relinquishing owners might not always be as truthful as expected [29], it is possible that this cat was infected by its owners, their relatives, or by (outdoor) feline contacts. The transmission of SARS-CoV-2 could, alternatively, have taken place by animal ambulance workers or shelter staff with COVID-19. Later, in group housing, there was also ample time for transmission through the direct contact with infected cospecies. Experimental studies have shown that the transmission of the virus between cats occur within two days of physical contact, and infected cats make antibodies within 10–14 days [1,2,6]. Although the female cat in this study showed an increase in titer between the two samplings, she remained the only seropositive case in this shelter. It therefore seems unlikely that this cat was infected during her stay in the shelter.

The seropositive stray male cat entered the shelter only three days prior to sampling. Being a stray cat, no information was available about his past. Considering the high antibody titer, and the fact that the cat was PCR negative when sampled three days after entering the shelter, we can assume that this cat had been infected prior to its shelter intake. In an experimental setting, cats infected by other cats turn SARS-CoV-2 RNA positive around day two post-contact and become negative again at day nine [2]. However, there is little knowledge about the transmission of SARS-CoV-2 between cats in real life. It therefore remains unclear when, and how, both seropositive cats became infected.

The study limitations should be considered when analyzing these data. Being a convenience sample, the population for this study has some biases. As sampling was not randomized, veterinarians might have preferred to sample cats which were easier to handle, or cats that had to undergo surgical procedures.

In the design of this study, a representative number of shelters were asked to participate. Several shelters declined the invitation for various reasons: some apprehended emotional stress among their volunteers in areas seriously affected by the pandemic, while others feared infection risks for their staff. Unfortunately, these shelters were not included.

During this pandemic, the public demand for pets increased substantially (based on personal communication with shelters). This had an unexpected effect on the time of sampling: some cats were sampled shortly after arrival, while others had been in the shelter for a longer time. Therefore, no conclusions about transmission routes within the facility could be made.

### Recommendations for Further Research

Although this study does not indicate that animal shelters are potential spillovers of SARS-CoV-2, either between pets or from pets to humans, continuous (sero-)surveillance within shelters remains important to monitor the possible changes in infection dynamics and the potential development of new virus mutants. Due to privacy regulations, information about COVID-19 infections among shelter personnel was unavailable for this study. This information should be included in future research, as it relates to the risks of anthroponotic infections within an animal shelter setting.

## Figures and Tables

**Figure 1 viruses-13-01634-f001:**
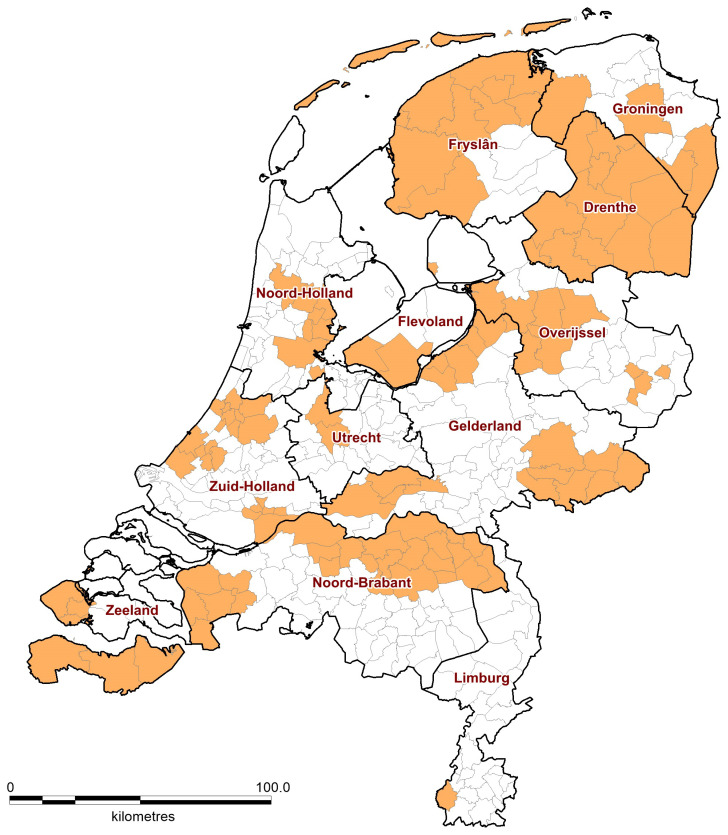
Outline of Dutch municipalities (apricot color) served by the participating animal shelters in this study.

**Table 1 viruses-13-01634-t001:** Signalment and management of 240 shelter cats from 28 Dutch shelters, sampled for serological prevalence of antibodies against SARS-CoV-2.

Gender	Total	Female	% ^2^	Male	% ^2^	Total	% ^2^
		109	45	131	55	240	100
Age category at sampling time	Kittens (0.5–<1 year)	11	10	17	13	28	12
	Young adult (1–<4 years)	52	48	48	37	100	42
	Mature adults (4–<7 years)	21	19	35	27	56	23
	Seniors (≥7 years)	25	23	31	24	56	23
			100		100		100
Origin	Stray	69	63	90	69	159	66
	Owner surrender	37	34	37	28	74	31
	Confiscated	3	3	4	3	7	3
			100		100		100
LOS ^1^ at sampling time	≤14 days	18	17	33	25	51	21
	>14 days	88	81	96	73	184	77
	Unknown	3	3	2	2	5	2
			100		100		100
Housed at sampling time	Solitary housed	32	29	52	40	84	35
	Group housed	72	66	79	60	151	63
	Unknown	5	5	0	0	5	2
			100		100		100

LOS ^1^ = Length of Stay, time between entry in the shelter and sampling time. % ^2^ = general distribution per gender, and per gender distribution per signalment/management.

## Data Availability

All data files will be available from the Yoda Data management system https://dgk.yoda.uu.nl/ (accessed on 25 March 2021) https://www.mdpi.com/ethics (accessed on 25 March 2021).

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
