# Peer review of "Serological Screening for Antibodies against SARS-CoV-2 in Dutch Shelter Cats"

_viruses, 2021, doi:10.3390/v13081634_

Round 1

Reviewer 1 Report

Taking into account the ‘One Health’ idea, to combat the COVID-19 pandemic, we should eliminate the SARS-CoV-2 not only from the human population, but also from the animal reservoir, i.e. cats. The recent data showed that the novel coronavirus can infect pet animals. The question whether the virus can be transmitted between them as well as between animal and human needs more data for the proper answer. Therefore, the results by Ruth van der Leij and colleagues are of high interest for all scientists fighting with  COVID-19. The manuscript is well-written and easy to follow. I have some minor comments, which can be found below.

MINOR COMMENTS

  • Please be consistent with the number spelling, all number > 10 should be written with numbers and all number <10 should be written with words, i.e. Page 2 Line 70.
  • It would be very helpful, especially for the readers outside the Netherlands, if you mark the 12 Dutch provinces, including the Noord-Brabant and Limburg, on the Figure 1.
  • Could you provide the information on the province’s origin for the 2 seropositive cats and can you compare this information with the COVID-19 confirmed cases in this region?

Reviewer 2 Report

The authors conducted an important work for screening SARS-CoV-2 antibodies in shelter cats. The result is valuable to figure out the infection status among cats and the potential transmission between pets and humans. There are two questions that should be considered before publishing.  

  1. Is it possible to discuss the probability of the cross-reactivity of the S protein, RBD and VNT between SARS-CoV-2 and FCoV? Were the positive results for antibody and VNT due to FCoV but not SARS-CoV-2?
  2. What kind of antibody was detected by ELISA? IgG or IgM? When would the oropharyngeal and rectal samples be taken after the seropositive results were obtained? If antibody is positive, it implies it’s a long time after infection and the virus usually is eliminated. So I don’t think PCR test is necessary and helpful after seropositive. It is highly meaningful to collect oropharyngeal and rectal samples from all the 240 cats and test nucleic acid when collect serum samples.
